# Stationed or Relocating: The Seesawing EMT/MET Determinants from Embryonic Development to Cancer Metastasis

**DOI:** 10.3390/biomedicines9091265

**Published:** 2021-09-18

**Authors:** Chien-Hsiu Li, Tai-I Hsu, Yu-Chan Chang, Ming-Hsien Chan, Pei-Jung Lu, Michael Hsiao

**Affiliations:** 1Genomics Research Center, Academia Sinica, Taipei 115, Taiwan; dicknivek@icloud.com (C.-H.L.); cardiosea@gmail.com (T.-I.H.); ahsien0718@gmail.com (M.-H.C.); 2Department of Biomedical Imaging and Radiological Sciences, National Yang Ming Chiao Tung University, Taipei 112, Taiwan; jameskobe0@gmail.com; 3Institute of Clinical Medicine, College of Medicine, National Cheng Kung University, Tainan 704, Taiwan; 4Clinical Medicine Research Center, College of Medicine, National Cheng Kung University Hospital, National Cheng Kung University, Tainan 704, Taiwan; 5Department of Biochemistry, Kaohsiung Medical University, Kaohsiung 807, Taiwan

**Keywords:** EMT, MET, embryonic, tissue repair, tumorigenesis

## Abstract

Epithelial and mesenchymal transition mechanisms continue to occur during the cell cycle and throughout human development from the embryo stage to death. In embryo development, epithelial-mesenchymal transition (EMT) can be divided into three essential steps. First, endoderm, mesoderm, and neural crest cells form, then the cells are subdivided, and finally, cardiac valve formation occurs. After the embryonic period, the human body will be subjected to ongoing mechanical stress or injury. The formation of a wound requires EMT to recruit fibroblasts to generate granulation tissues, repair the wound and re-create an intact skin barrier. However, once cells transform into a malignant tumor, the tumor cells acquire the characteristic of immortality. Local cell growth with no growth inhibition creates a solid tumor. If the tumor cannot obtain enough nutrition in situ, the tumor cells will undergo EMT and invade the basal membrane of nearby blood vessels. The tumor cells are transported through the bloodstream to secondary sites and then begin to form colonies and undergo reverse EMT, the so-called “mesenchymal-epithelial transition (MET).” This dynamic change involves cell morphology, environmental conditions, and external stimuli. Therefore, in this manuscript, the similarities and differences between EMT and MET will be dissected from embryonic development to the stage of cancer metastasis.

## 1. Introduction

The embryonic development of metazoan organisms starts from a single layer of cells [1]. These single-layer cells can be induced into pluripotent stem cells containing endoderm, mesoderm, and ectoderm [2]. These layers contain epithelial cells that play essential roles in organ development, cell reprogramming, tissue damage repair, and cell mobility [3,4,5,6,7,8,9]. Under typical situations, growing cells always maintain tight junctions, adhesion, cell–cell adhesion, and polarity [10]. Especially in the lung or small intestine, respiratory cilia or intestinal villi can help organs perform the correct function [11,12,13]. During development, epithelial cells undergo mesoderm formation [7], neural crest formation [14,15,16], cardiac valve formation [17], secondary platelet formation, and somitogenesis [18], during which EMT progression is necessary to create the mesoderm, neural crest, and heart valve, and promote male Müllerian duct progression [19,20]. If there are any mistakes during development, embryonic hypoplasia is observed. Therefore, identifying the key factors for the dominant cell’s pluripotency has always been an important issue. Scientists have discovered that mature epithelium could be induced to form pluripotent stem cells by overexpressing Oct4, Klf4, Sox2, and c-Myc [21,22]. However, the success rate of this process has not yet achieved its goals for use in humans. Additionally, a similar processes of EMT can be observed when internal organ damage occurs [23], and some epithelium undergoes trans-differentiation to repair the damaged part [24]. When tissues are wounded, these phenomena can easily be observed in the whole body or on the skin or body surface, where epithelial trans-differentiation into mesenchymal cells with cell mobility occurs [25]. Mesenchymal cells come to the wound region to form an intact barrier and transform into epithelial cells during the “wound healing process” [26]. Interestingly, these EMT-related pluripotency genes are controlled strictly in differentiated cells, and they can be found to be overexpressed in dysregulated cells, such as tumor cells, and usually accompanied by EMT and stem-like ability [27]. In addition, not only in embryogenesis, the most frequently mentioned EMT in recent years is the critical process associated with tumor metastasis [28,29]. Tumor metastasis requires tumor stemness ability and EMT mobility [30]. Then, cells can grow quickly and take host nutrition. However, the primary site of the tumor origin is usually accompanied by mass necrosis or apoptosis that triggers tumor cells to migrate from the original site to seek secondary sites to find nutrition, which is called the metastasis process [31], during which the tumor cells can break the extracellular matrix and cross the epithelium barrier, then extravasate into the blood vessels to form a colony in the endothelial cells [32]. Once the colony is formed by intravasation into the secondary site, it generates a secondary tumor, which may lead to cancer patient death [33,34]. Therefore, this review used embryogenesis and cancer metastasis as a biological model to discuss the related events on epithelial and mesenchymal transition.

## 2. The Principle of EMT and MET

Under normal circumstances, epithelial cells will keep forming a barrier or separate space [35]. Under the skin tissues or when the barrier has been breached, the epithelium transforms into fibroblast-like cells, which can be recognized by their cell morphology [36]. The epithelium forms flat and round cells. However, fibroblast-like cells look like a spindle with a lamellipodia and filopodia structure [37]. During this transition, epithelial cells transform into mesenchymal cells and gain the ability to move [38]. In addition, the wound regions secrete cytokines and inflammatory substrates to attract immune and mesenchymal cells for repair [39,40]. The processes can be dissected into the first step, whereby mesenchymal cells begin to form the wound edge, and then the cells will move forward into the central part to fill out the damaged part. The cells concentrated at the site of damage undergo epithelial-mesenchymal transition during the wound repair [41]. Unlike normal cells, during MET progression, cells lose their tight junctions and polarity, accompanied by increased CD44v and E-cadherin expression to maintain an epithelial phenotype [42]. Under the EMT process of the cell, motility can be confirmed by mesenchymal-related markers, such as CD44s, N-cadherin, discoidin domain receptor 2 (DDR2), β-catenin, vimentin, and α-Smooth Muscle Actin (α-SMA) [43]. A model EMT/MET transition is shown in Figure 1. Furthermore, EMT and MET are an essential part of the wound healing process [44] and create a cycle used for maintaining tissue volume [45,46,47,48].

## 3. EMT and MET during Embryonic Development

In embryonic development [49], the processes involving EMT are somitogenesis, nephrogenesis, carcinogenesis, cardiogenesis, and foregut development [18,50,51]. There are three steps of EMT: primary, secondary, and tertiary processes [52]. Primary EMT, including the formation of parietal endoderm, mesoderm, and neural crest delamination [53,54,55], and similar signaling occurs during gastrulation and neural crest formation [49]. A high degree of cooperation in the neural crest makes it plasticized during the primary EMT. Secondary EMT mesodermal cells are subdivided [56] and somitogenic [46,57,58], and endocrine cells migrate to the mesenchyme during pancreatic and hepatoblast formation, which plays an important role in platelet and reproductive tract development [3]. The tertiary step is cardiac valve formation [59,60,61]. In addition, metazoan formation induced by Snail1/2 is crucial during vertebrate head development [62]. VE-cadherin is suppressed by Snail1, and inactivation of Hey2, Hey1, and Heyl induces major congenital heart defects, combined with inactivation-induced ventricular septal and pulmonary vale defects [63] (Figure 2).

## 4. EMT in Tissue Repair

In a regular situation, tissue can be damaged by a wound [64]. Many steps are included in the wound healing process, such as hemostasis, inflammation, and remodeling of the injured tissue [65]. These steps include numerous cell types, including macrophages, keratinocytes, fibroblasts, platelets, and endothelial cells [66]. The related signaling pathways include the epidermal growth factor (EGF), transforming growth factor-β (TGF-β), fibroblast growth factor (FGF), and hepatocyte growth factor (HGF) signaling pathways [67]. Once the skin barrier is breached, it creates a wound, and EMT starts. Cells begin migration, intracellular dissociation, and matrix degradation and trigger the release of FGF, EGF, and TGF-β, as well as the HGF signaling pathway, to heal the wound [68]. Reepithelization is the process by which keratinocytes migrate from the wound edge to the central region [69]. A granulation site is observed under the wound center. The keratinocytes around the edge can separate and be individually transformed from flat, adherent cells into spindle-like cells [70]. Cells with a mesenchymal-like phenotype gain motility and dissociation and move to the central zone during wound healing [71]. The tissue can be healed by the moving cells, maintain homeostasis, and then finally transform back into epithelial cells [72] (Figure 3).

## 5. Differentiation and De-Differentiation between EMT and MET

During embryogenesis, the processes of EMT allow pluripotent cell movement to exact position within an appropriate time to undergo cells’ specification and differentiation, which would be a process of differentiation potential loss [38]. In addition to the cells’ differentiation, de-differentiation has also been involved in repairing cell stress, and is mediated by the EMT or MET [3,73]. Such processes can be easily found in the transition of myofibroblasts and fibroblasts [25]. Multiple cell resources have been identified to be able to transform into myofibroblasts under cell stress or injury. The differentiated myofibroblast can be formatted from the differentiation of the fibroblast, de-differentiation of smooth muscle cells, and EMT transformation of the epithelium or endothelium [74]. Even the smooth muscle, adipocyte, and pericyte can trans-differentiate into the cancer-associated fibroblast [75]. The TGF-β1 has been considered as a dominant factor of related events by regulating the ALK5/Smad3 axis [76,77]. Tissue injuries, such as exposure to a radiation environment, can quickly induce the expression of TGF-β1 [78], which results in extracellular matrix deposition and drives EMT-mediated fibrosis processes [79]. Related markers, such as FSP1, α-SMA, and collagen 1, have been used [80,81,82]. Mathison et al. showed that the artificial manipulation of the EMT-related axis, GATA4/SNAIL, could reduce cellular differentiation in cardiac fibrosis [83]. In addition, overexpression-specific factors, including GATA4, FOXA3, HNF1A, and HNF4A, can reprogram myofibroblasts into hepatocytes [84]. A similar phenomenon was observed in cancers. The tumor cells can increase the population of myofibroblasts by secreting TGF-β to transform surrounding cells components, such as the cancer-associated fibroblast [85,86], and consequently promoting the EMT and chemoresistance. Moreover, TGF-β was found to regulate cells from an epithelial phenotype into partial-EMT or hybrid EMT cells, which were found with multiple features under this heterogeneous transient population, such as collective migration or stem-like abilities [87]. Interestingly, under this partial-EMT transient status, cells can maintain epithelial markers, but also with mesenchymal ability, such as stem-like and collective motility, as compared to epithelial status, consequently promoting survival under diverse strict conditions or metastasis ability [87]. Most importantly, in addition to the related biological process, the differentiation and de-differentiation status, as well as related markers, can be applied for clinical diagnosis [88,89].

## 6. EMT at the Primary Tumor Site

### 6.1. Primary Tumor

Regular cells transformed into malignant cells which proliferate more than normal cells [90]. The cancer niche contains cancer-associated fibroblasts, cancer cells, tumor-associated macrophages, extracellular matrix, and endothelium [91,92]. The primary tumor will secrete Vascular endothelial growth factor (VEGF) to attract endothelial cell migration to the tumor to supply oxygen and nutrition [93,94]. In addition, microvessels can also increase the metabolism rate and enrich the tumor growth [95] (Figure 4).

### 6.2. Primary Tumor Extravasation

Once the tumor cells’ gain of proliferation ability can boost growth quickly, the size of the primary tumor increasing to more than 1 cm^3^ can induce tumor central zone necrosis and hypoxia and then release cytokines [96,97]. In the primary tumor site, tumor cells have E-cadherin, tight junctions, cell adhesion junctions, and desmosomes [98,99]. When cells undergo the EMT process, the cells experience cytoskeletal reorganization [100]. The extracellular matrix is degraded with extra metalloproteinase (MMP) secretion, and the cells can invade the basal membrane [101]. Epithelial markers such as ZO-1 or E-cadherin are lost [102]. Mesenchymal markers and transcription factors are expressed [57,58,103]. These cells migrate individually or coordinate together and migrate in a similar direction at the same speed to invade the basal membrane near vessels [104]. The primary tumor can also be recruited to microvessels, causing tumor cells to invade the circulation [105] (Figure 5).

## 7. EMT in the Circulation System

During circulation, tumor cells can be induced by platelet-produced TGF-β/Smad [106,107,108], and insufficient nuclear factor kappa-light-chain-enhancer of activated B cells (NF-κB) promotes cell adhesion and extravasation [109]. Migrating cells adhere to the endothelial cell wall in vessels, and the cells then circulate throughout the body [110]. The blood flow across the cells can create a sheath force, and the cell cannot attach to the surface without adhesion [111,112], so the cells cannot sustain growth and proliferation [113] unless they can attach to the endothelium cell wall and recruit other cells to form a cell plaque, including fibroblasts, endothelium cells, and myeloid progenitor cells [114,115]. These initiated cells will form tumor lesions in vessels and grow in the secondary site [116] (Figure 6).

## 8. EMT in the Metastatic Site

On the front edge of EMT, cells can develop intravasation into secondary regions [117]. Some cancer stem cell theories call it a niche, while others indicate that the cells can form a plaque and increase the cells’ intravasation into the basal membrane to create a secondary metastasis [89]. In previous research, cancer-associated fibroblasts, such as bone marrow-derived mesenchymal stem cells, have been identified [118]. The other question is whether the primary migrating cells with invasion gain other abilities, like cancer stem cells transforming into vessels or lymphatic cells to promote cells with more aggressive motility [119]. Invaded cells in secondary sites and secondary metastasis represent the most common reasons for cancer death and are not easily identified [115,116,120].

## 9. Colonization between EMT and MET

The invading cells migrating into the secondary site will require MET to colonize the metastatic region [121]. Spindle-like cells transform into epithelium-like cells and lose mobility, which is called MET [122]. The cells can attach to the organ or cells with cytoskeleton changes allowing for expansion and outgrowth, which is necessary for colonization [123]. Nevertheless, it has also been demonstrated that Twist-1 is upregulated in EMT and after colonization and Twist downregulation is associated with stemness ability, but it does not activate proliferation [124]. The other type is the Paired Related Homeobox 1 (Prrx-1) type, in which EMT activates Prxx-1 activity but suppresses stemness ability. After colonization, Prxx-1 is downregulated and activates stemness to promote colonization [125] (Figure 7). Cancer stem cells are embedded among tumor cells [126]. However, these findings indicate that the MET process is the most crucial issue in colonization. Table 1 lists the genes that are related to EMT and MET.

## 10. Defined EMT Molecules in Cancer

At present, the critical EMT or EMT molecules discovered in cancer have been identified [127] (Table 2 and Table 3). These molecules participate in cancer’s tumor-initiating ability, invasiveness, stemness, and drug resistance [128,129,130]. Of note, they also have other biological functions [131]. Interestingly, these molecules are involved in cancer-initiating ability, invasiveness, stemness, and drug resistance. These established molecules can be seen to have biological functions of adhesion of embryonic cells by in silico simulation assays (Figure 8). In addition, these molecules participate in many essential canonical pathways related to cancer progression. Therefore, these common molecules can not only serve as prognostic markers, but they also activate signaling pathways and can be used to understand how cancer cells convert from the epithelial type to the mesenchymal type. The following correlations are proposed based on the simulated results shown in Table 2 and Table 3.

### 10.1. Cytoskeleton Remodeling

Transformation of the epithelial-mesenchymal system can trigger the remodeling of cell types [132]. The signals involved in EMT include Rho-regulated actin motility signaling, and Rho GTPase signaling. These types of changes mainly regulate the binding, polymerization, and stabilization processes to reorganize the cytoskeleton made up of actin [133,134]. RhoGDP-Dissociation Inhibitor α (RhoGDIα) is an essential negative regulator that interacts with small GTP-binding proteins, such as RhoA, Cell Division Cycle 42 (CDC42), and Rac1, to control cell migration and it is responsible for the formation of cells with multiple morphologies, such as membrane protrusions, lamellipodia, and stress fibers [135]. Dysregulation of RhoGDIα is a prognostic marker in cancer, and it has been found to be overexpressed in diverse cancer types, such as colorectal cancer and nasopharyngeal carcinoma [136,137]. Overexpression of RhoGDIα can promote lung metastasis in bladder cancer [138]. Loss of the interaction between RhoGDIα and CDC42 increases tumor metastasis [139]. A similar study reported that Ephrin B2 can stimulate the interaction of RhoGDI and Ephrin B1, and then RhoA is released from RhoGDI to facilitate cell migration [140]. Conversely, increased RhoGDI levels can also be observed to decrease the stemness ability of tumor cells [141]. In addition, RhoGDI can be regulated by interferon-gamma (IFN-γ) [142]. Therefore, it is predictable that targeting RhoGDI can modulate tumor cell motility.

Integrin-mediated focal adhesion is the general way to control actin polymerization [143,144,145]. Integrin has been reported to participate in cell motility through related signaling pathways, such as Rho-regulated actin motility signaling, and Rho GTPase signaling [146,147]. For example, Integrin Subunit Alpha 2 (ITGA2) was found to affect cell motility in esophageal squamous cell carcinoma by regulating the focal adhesion kinase/protein kinase B (FAK/AKT) axis [148]. Integrin α5 can regulate migration and invasion through the FAK/STAT3/AKT axis, and it also contributes to cell resistance to chemotherapy-related treatment [149]. Moreover, multiple molecules can be observed to regulate cells by crosstalk integrin-related signaling, such as Myosin Heavy Chain 9 (MYH9), Rab11, and Wnt5a [150,151,152], especially the latter, as the paracrine factors involved in EMT/MET have been found to modulate the ITGAV level in ovarian cancer [152]. Interestingly, the interaction of osteopontin and integrin αvβ3 can accelerate cellular uptake of glucose to improve cell migration ability [153]. A similar study showed that the interaction of these molecules could increase the resistance of lung tumor cells to gefitinib treatment [154]. This indicates that the noncanonical pathway of integrin signaling contributes to diverse biological functions and it requires more comprehensive exploration in cancers.

Additional related stimulation factors of cells include growth factors, extracellular matrix, and G-protein-coupled receptor (GPCR), in which the GPCR is a transmembrane protein with rich G proteins, including Gs, Gi, Gq, and G12/13, that can regulate cell functions through adenylyl cyclase, phospholipase Cβ, and the Rho family [155]. Yifan Wang et al. showed that the expression of protease-activated receptor 1 (PAR1) was correlated with a poor prognosis of ER-negative breast cancer, in which PAR can regulate the YAP/TAZ axis to increase cell motility and cancer stem-like activity [156]. In hepatocellular carcinoma, increased GPCR kinase interacting protein-1 (GIT1) in tumor tissues is associated with a poor survival rate. However, in contrast to the canonical pathway, GIT1 can regulate cell motility through ERK signaling [157]. A similar phenomenon was observed in the Angad Rao et al. study, in which overexpression of GPR19 through the ERK axis modulated cell migration and invasion ability [158]. However, some members play tumor suppressor roles. In nasopharyngeal carcinoma, low apelin receptor (APLNR) expression is correlated with a poor prognosis, and knockdown of APLNR can improve cell motility [159]. Similarly, Aleena K S Arakaki et al. showed that α-arrestin domain-containing protein 3 (ARRDC3) could affect tumor cell metastasis by modulating the Hippo pathway [160]. Therefore, multiple inhibitors have been developed to restrict GPCR activity. Sunitinib, a kinase inhibitor, was found to inhibit downstream CDC42 and RHO kinases 1 (ROCK1) functions by targeting G Protein-Coupled Receptor Kinase 5 (GRK5) to modulate the motility of triple-negative breast cancer cells [161]. Jan Stein et al. found that teleocidin A2 can inhibit proteinase-activated receptor 2 (PAR2) in breast cancer [162]. Similarly, a coenzyme Q10 analog, decylubiquinone, was found to inhibit brain angiogenesis inhibitor 1 in breast cancer, thus inhibiting metastasis [163]. Natural products such as yohimbine have also been found to inhibit the activity of arginine vasopressin receptor 2 [164]. In addition, the adenosine A2B receptor was found to have diverse mutation points. Xuesong Wang et al. identified different selective inhibitors, such as NECA, BAY 60-6583, ZM241385, and PSB603 [165].

Actin filament reorganization, which induces axon attraction, repulsion, and outgrowth, is an EMT process that also participates in axonal guidance signaling [166,167]. Multiple exon guidance-related molecules have also been found involvement in cancer progression [168]. Artificial manipulation the expression of Neogenin-1 was found to affect neuroblastoma metastasis [169]. Molecules related to neuronal migration, such as semaphorin 4C, have also been found to regulate cell motility in breast cancer [170]. Similar functions were also found by Smeester et al., who showed that semaphorin 4C can regulate the migration of osteosarcoma to the lung by regulating p-AKT activity [171]. Additionally, secreted molecules that play an essential role in developing cortical neurons, such as Netrin-1, have been found to regulate the ERK/FAK axis in colorectal cancer to control the axon outgrowth of cells [172]. However, in ovarian cancer, Netrin-1 was found to be a tumor suppressor, modulating BMP signaling [173]. Interestingly, the overexpression of Netrin-1-induced axonal guidance can suppress BMP4 expression, which was involved in MET processes, indicated that Netrin-1 could dominate both the EMT and MET [173]. Similar functions can also be seen for semaphorin 3B, but it functions only in nonneuronal cells. For example, in breast cancer, semaphorin 3B inhibits GATA3 to reduce cell migration [174]. In HER2-related drug-resistant breast cancer cells, it was also found that an increase in neuropilin-1 can benefit cells, bypass drug inhibition, and promote cell growth in the lung [175]. In addition, these stimuli are also accompanied by the activation of ILK signaling, resulting in cell type changes that allow the cells to migrate [176,177,178].

### 10.2. EMT to Metastasis

EMT regulation by growth factors and others also showed that extrinsic factors, including IL-6, Tnf, TGF-b, Notch, Wnt, and receptor tyrosine kinase EGF, HGF, PDGF, FGF, etc., can further enhance cell invasion ability and facilitate cells to secrete matrix metalloproteinases (MMPs) to digest ECM [179,180,181]. Currently, approximately 23 MMP members have been identified [181]. In cancer, MMP2 and MMP9, which are the most commonly discussed, are classified as gelatinases (A and B) [182]. They are also regulated by many EMT-related molecules, which in turn express and promote tumor invasion and metastasis [183]. Interestingly, cancer cells are also stimulated by ECM to activate the GP6 signaling pathway and integrin signaling [184]. GP6, also known as glycoprotein VI or GPVI, is activated by many factors, such as extracellular matrix, including fibrinogen, laminin, CRP, collagen, convulxin, and alborhagin. It is also found mainly in platelet plasma membrane proteins [185]. Therefore, many studies have explored the role of the interaction between platelets and tumors in tumor progression [186]. Mammadova-Bach et al. showed that the interactions between GPVI in platelets and galectin-3 on tumor cells could promote tumor cell extravasation, thereby inducing cell metastasis [184]. However, targeting GPVI may cause disadvantageous side effects. According to the report by Tullemans et al., the use of pazopanib, a selected tyrosine kinase inhibitor, can cause mild bleeding in patients with renal cell carcinoma. The main reason may be that GPVI inhibition can cause Ca^2+^ to increase and inhibit the function of phospholipid phosphatidylserine [187]. The calcium level was associated with the actin polymerization and cell motility [188,189], in which Wnt5a as a ligand of the Wnt-Ca^2+^ non-canonical pathway drives the calcium release to modulate downstream PKC, calcineurin, and CaMK-II activity [190,191]. This is similar to the vicious cycle that allows cancer cells to benefit from cell-cell interactions. Eventually, it is possible to see the activation of colorectal cancer metastasis signaling and glioma invasiveness signaling in simulated results.

### 10.3. Remodeling the Microenvironment

The oxygen concentration determines the survival of cells [192]. Unlike regular cells, cancer cells can survive in a low-oxygen environment [193]. The EMT can be related to the activation of HIF-1α signaling, which promotes cell motility, contributes to the nutrient metabolism of cells, and activates angiogenesis to increase the energy supply so that cancer cells can survive in a harsh environment [194]. Otto Warburg proposed the relationship between Hif-1α and cancer in 1923 as the “Warburg effect” to explain cells surviving under hypoxic conditions. Hif-1α can act as a transcription factor to transactivate downstream effectors to accumulate CUL4B levels in breast cancer, consequently suppressing epithelial-related factors, such as E-cadherin and AXIN2, by cooperating with ZEB2 and Snail to promote metastasis [195]. In addition, cells in an environment of hypoxia have changes in their activity of metabolism-related enzymes [196]. In lung cancer, ALDOA, one of the glycolysis- and gluconeogenesis-related enzymes, is activated by HIF-1α in a hypoxic environment and it promotes lactate increases, consequently prolonging the protein half-life of HIF-1α and forming a circuit regulation [197]. In addition to stimulating growth factors, inflammation such as IL-17 signaling can also crosstalk with HIF-1α signaling [198]. Recently, studies have found that the interleukin-17 receptor B is positively associated with the degree of migration [199]. In pancreatic cancer, the phosphorylation of tyrosine at position 447 on the interleukin-17 receptor B can be used as a prognostic marker, promoting metastasis when the corresponding ligand, IL17B, stimulates cells [200].

Once the cells extravasate to the outside of the tissues or organs, the tumor microenvironment is remodeled [201]. The most common cells surrounding tumor cells are stromal cells, immune cells, mesenchymal stem cells, endothelial cells, adipocytes, and fibroblasts. Recently, many stromal cells have been shown to be assimilated by tumors [202,203]. As a result, these cells are redefined by new terms, such as tumor-associated neutrophils (TANs), tumor-associated macrophages (TAMs), cancer-associated fibroblasts (CAFs), tumor-associated bone marrow-mesenchymal stromal cells (BM-MSCs), and tumor-associated adipose tissue-mesenchymal stromal cells (AT-MSCs) [204]. EMT can also affect hepatic fibrosis/hepatic stellate cell activation and signaling pathways, and tumor microenvironment pathways [205]. This shows that tumor cells can create a distinct environment, such as acidification. Fibroblasts are the most common extracellular stromal cells. Once stressed, they will be activated and transform into the mesenchymal type or myofibroblasts. The sources of activated fibroblasts may be local fibroblasts, vascular pericytes, bone marrow-derived cells, endothelial cells, epithelial cells, etc. [206]. 

Recently, cancer-associated fibroblasts have also been found to be complementary to cancer, helping tumors become resistant to drug treatment [207,208]. Recurrence caused by drug resistance is currently one of the major factors leading to the failure of many drugs, and Apicella reported the failure of chemotherapeutic drugs mediated by CAFs [209]. In this exciting project, it was first discovered that the content of cell metabolites, the lactate content, and the resistance of cells to tyrosine kinase inhibitors, such as MET/EGFR-related drugs, have a remarkable correlation. Despite its current clinical use, the drug causes high-intensity damage to only a particular population of tumor cells. However, due to the heterogeneity among cells, the tumor will gradually become resistant during treatment, and this will lead to recurrence. Interestingly, when isolated drug-resistant cells are repeatedly treated, similar results are obtained. Finally, the drug resistance of cells is mainly derived from the communication between CAFs and tumor cells, in which tumor cells that produce lactate will cause CAFs to feedback growth factors, such as HGF, to strengthen cell resistance to drug treatment [209]. However, these complex interactions need more research to clarify, and whether they are appropriate therapeutic targets is still unknown. Additionally, CAFs and tumor cells interaction can switch the plasticity and heterogeneity of cells. For example, Mosa et al. found that tumor cells and CAFs interaction drives high αSMA expressed in CAFs and secreted a high level of Wnt to promote tumor growth, with the Wnt level as a switch factor, once interference with Wnt signaling can drive more inflammatory-like CAFs and consequently promote EMT formation [210]. Therefore, identifying the critical factors for the dominant cell plasticity may uncover how to switch between the EMT and EMT.

### 10.4. Upstream Regulators Contribute to Determining the EMT/MET in Cancers

The simulated analysis revealed the possible upstream factors involved in the EMT and MET switch (Table 3). Their gene ontology is linked to both embryogenesis and cancer progression (Figure 9), which are involved in well-known signaling, such as TGF-β, BMP, growth factors, Wnt, Notch, Hedgehog, Integrin, and Hippo [211]. The simulated results showed that both EMT or MET link to the cell differentiation and cell motility pathway. These relationships match the criteria that cells switch the status between epithelial and mesenchymal to possess related biological functions (Figure 9). During development, mesenchymal progenitor disorder can cause divers syndromes, such as Job syndrome, caused by abnormal immune or skeleton development, the reason for which is the mutation of STAT3 resulting in the loss of Wnt/β-catenin-related transduction, and can be restored by GSK3 inhibitors [212]. Lin et al. found that different position phosphorylation on STAT3 can switch the epithelial and mesenchymal to control cancer cell motility [213]. 

Moreover, to move to the correct position to conduct sub-population differentiation, the progenitor cells can modulate the E-cadherin to upregulate SNAIL1/2 resulting in partial EMT [214]. The expression of SNAIL1 can serve as an EMT hallmark in cell biology, especially in cancer progression. The expression of SNAIL1 can be regulated by the STAT3/LIV-1 axis. This regulation controls the anterior migration of gastrulation in embryogenesis [215] and contributes to the tumor invasion, stem-like motility and chemoresistance [216]. In addition, not only SNAIL1, the related EMT transcription factors such as ZEB1 and twist1 were also found to be regulated by MET-related dominant molecules, the OVOL1/2 [217,218,219], and GRHL2 [220,221], and such factors were involved in wound healing and embryogenesis as well [222,223,224]. According to the report by Pienta et al., both OVOL1/2 serve as negative regulators in prostate or breast cancer, in which overexpression of OVOL1 or OVOL2 can decrease ZEB1 to restrict cells EMT, consequently suppressing metastasis [225]. Similar observations were described in another MET-related factor, the GRHL2. In carcinoma or breast cancer, GRHL2 can suppress ZEB1 to block TGF-β-mediated EMT and resistance ability [226]. Such reciprocal regulation may contribute to the promoter binding regions. Cieply et al. demonstrated that GRHL2 could directly bind to the ZEB1 promoter, and the interaction of GRHL2 and Six1 may accelerate specific recognition for its promoter binding [226,227]. However, the controversial roles of GRHL2 have been described in breast cancer, in which overexpression of GRHL2 drives EMT phenotype via binding to ERBB3 promoter [228]. In agreement with other reports, Faddaoui et al. found that GRHL2 was associated with cell growth and motility phenotype in ovarian cancer [229]. Interestingly, ZEB1 can bind to the GRHL2 promoter as well [227]. Similarly, this regulation can be observed in the promoter regions of ZEB1 or twist1, which were with the OVOL2 binding sites [225,230]. Therefore, such regulatory circuits provide cells plasticity between the EMT and MET. Additionally, the epithelial-related molecule E-cadherin has been considering as a β-catenin limiter. Under Wnt ligand stimulation, β-catenin can translocate into the nucleus as a transcription factor to activate the TCF/LEF axis [231]. Factors including TGF-β, growth factors, and Hedgehog mediate upstream switch factors, such as Smad3/4, JUN, ETS1, NF-κB, and GLI1/2/3, with a similar phenomenon regulation between the EMT and MET [232,233,234,235,236]. Furthermore, as transcription modifiers, such as the HOTAIR regulatory pathway, long-noncoding RNA was found to link the development and cancer [237]. All the possible networks are described in Figure 9 and Table 3. Interestingly, most of these critical switch factors are involved in pluripotent features.

### 10.5. Stemness and EMT/MET

Stemness is an intermediate transition status to link reversible differentiation and de-differentiation, which provide a sub-population of embryonic cells that undergo EMT and MET to develop specific cells and can also be formed from the EMT of epithelial cells or differentiated cells [73]. In cancer, stem-like features empower cells’ plasticity and are considered a reason for the failure of chemotherapy [238,239]. The related stemness factors can be transactivated under the transient EMT/MET processes, consequently promoting the motility of cells and surviving in rigorous environments [240,241]. Subjected to this switch, upstream factors between EMT and MET can be linked to self-renewal signaling (Figure 9), the features of which can be observed in the partial EMT; similar heterogeneous populations have been identified with unusual EMT characteristics, stemness and drug resistance, consequently stimulating metastasis events [242,243,244]. The related stemness factors can be observed in Table 3. There are multiple signaling molecules involved in cell pluripotency and stemness, such as Sox2, Oct4, and Nanog, as the key pluripotency-related genes [245], in which their expression activity can be regulated by EMT/MET-dominant factors, including STAT3, Smad3/4 and β-catenin, to control the Jak–STAT, TGF-β, and Wnt signaling pathways. In addition, these regulations contribute to ectoderm development and the transcriptional regulatory network in embryonic stem cells, with related factors, such as Pax6, Meis1, Lhx5, Otx1, and Neurog1. Mainly, this type of stem-like cell has an unusual cell cycle compared to epithelial cells, and the related genes include RB1, SMAD, Myc, and TP53. Myc is an essential transcription factor and is regulated by STAT3 and β-catenin, which are involved in both development and tumorigenesis [246]. Das et al. found that the related regulation of HIF2α, as a down-stream effector of Myc, can cause the cancer stem cells to undergo self-renewal or differentiation, in which the interaction among Myc, Sox2, and Nanog on the HIF2α promoter can modulate the level of p53, Glutathione (GSH) and reactive oxygen species (ROS) to determine cell fate [247]. In addition, the overexpression of Myc can induce EMT-related genes, such as SNAIL, ZEB, Twist, OPN and SALL4 [248]. These regulations contributed to the partial EMT, drug-resistant or immune escape abilities, consequently causing a relapse of cancer. Therefore, the identification of the essential dominant stem-like factors may provide alternative targeting strategies for cancer progression.

## 11. Conclusions

To the best of our knowledge, EMT and MET are essential to cancer [126]. Therefore, cells that move and gain malignancy are important issues that need to be investigated. However, under both regular and malignant conditions, EMT and MET always occur [249]. The molecular mechanism involves FGF, TGF-β, STAT3, and ER under normal conditions to repair wounds. A similar molecular mechanism is also used by tumor cells to help tumor cells occupy different organs and obtain nutrition for cell survival (Figure 10). Therefore, understanding the molecular mechanism in tumors may identify significant therapeutic targets for cancer therapy [250].

## Figures and Tables

**Figure 1 biomedicines-09-01265-f001:**
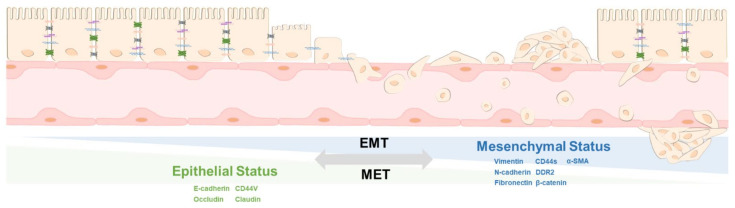
The regulation of EMT and MET. Epithelial cells can transform the original cell type from epithelial status to mesenchymal status through the EMT process, and its ability to move, such as entering the circulation system and then returning to epithelial status, through the conversion of MET. Therefore, the level of EMT and MET mobility and its corresponding markers to distinguish the types of EMT and MET is described.

**Figure 2 biomedicines-09-01265-f002:**
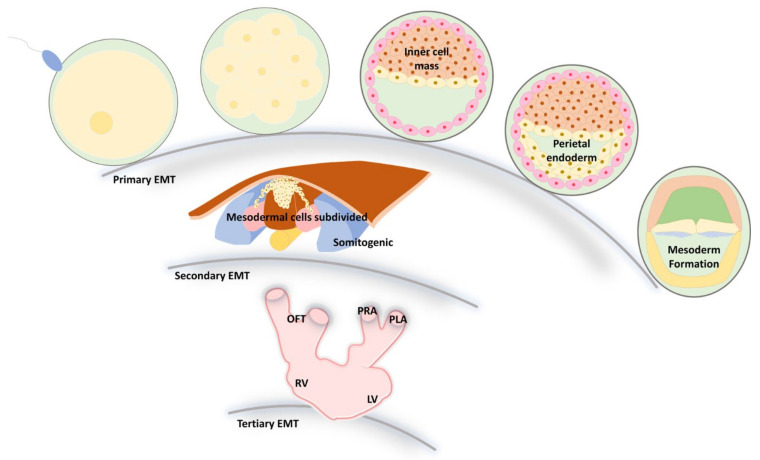
EMT progression of development in the first, secondary, and tertiary embryo stages. EMT plays an essential role in embryonic development and is closely related to the movement of different germ layers to specific locations for further differentiation. During embryogenesis, the embryo can undergo three major EMT processes. First, once the zygote is formed, the cells will move to a specific location to distribute three primary germ layers, endoderm, mesoderm, and ectoderm, called primary EMT. The second EMT can be observed in the epithelial structure among the mesoderm, notochord, neural tube, and somite, which undergo processes such as endocrine cell formation. Finally, the formation of mesenchymal cells, such as cardiac cushions as cardiac valve precursors, is the most appropriate model to describe tertiary EMT. OFT  =  Outflow Tract; RV  =  Right Ventricle; PRA  =  Primitive Right Atrium; PLA  =  Primitive Left Atrium; LV  =  Left Ventricle.

**Figure 3 biomedicines-09-01265-f003:**
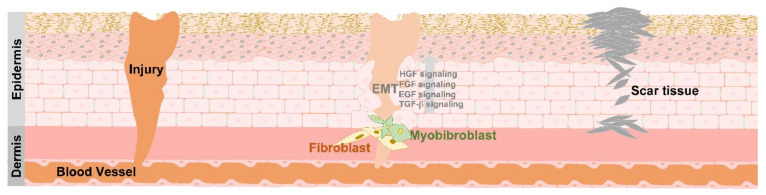
Wound healing progression. The diagram shows wound healing progression. Different signal transmissions interact with neighboring cells through EMT when the tissue is damaged to repair the wound. The related signaling, such as FGF, EGF, TGF-b, and HGF, can induce cell migration by redefined epithelial, mesenchymal, and secret matrix metalloproteinase to remodel extracellular components. In addition, cells, such as Keratinocyte and fibroblast, can undergo the EMT and cellular proliferation process to participate in wound healing-related processes.

**Figure 4 biomedicines-09-01265-f004:**
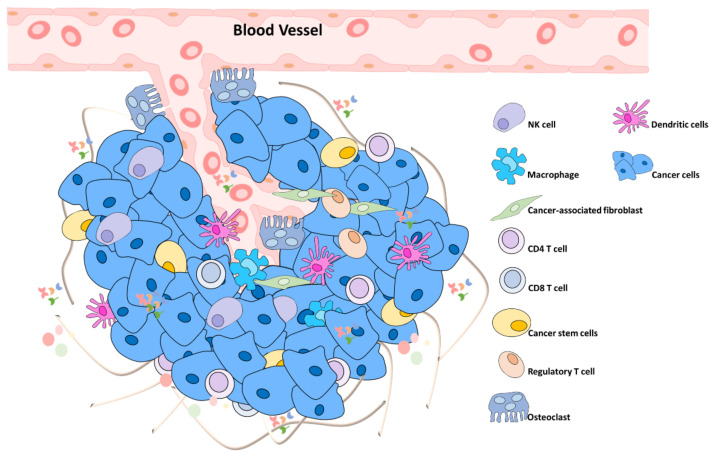
Primary tumor. The tumor microenvironment contains tumor stem cells and tumor-associated extracellular components, such as immune cells. Mainly, cancer cells, such as tumor-associated macrophages, tumor-associated adipocytes, tumor endothelial cells, and tumor-associated fibroblast, can change the characteristics of neighboring cells in various ways and further interact with each other to increase the degree of tumor heterogeneity.

**Figure 5 biomedicines-09-01265-f005:**
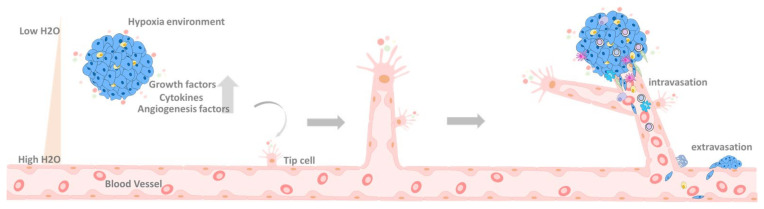
The progression of extravasation. Cancer cells can migrate from carcinoma in situ through blood vessels to other parts of the body through the extravasation process. Under this process, the hypoxia environment can stimulate tumor cells to secret chemotactic factors, such as growth factors, cytokines, and angiogenesis-related factors, stimulating epithelial cells for blood vessel interaction through angiogenesis. In such cases, tumor cells will develop intravasation into the blood vessel and the extravasation process to uptake more nutrients. Up arrow means upregulation.

**Figure 6 biomedicines-09-01265-f006:**
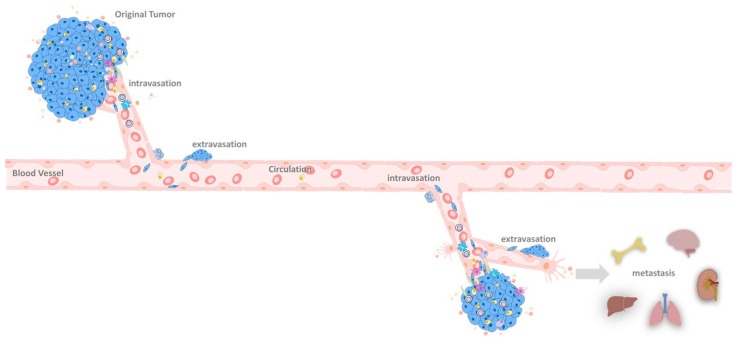
Metastasis progression. The illustration shows how cancer cells obtain more nutrients through angiogenesis and grow outwards. Through extravasation, they can transfer to other parts of the body for further expansion. The original tumor cells can undergo EMT and intravasation and circulate in blood vessels. Under the EMT process, cells survive in strict circumstances and through extravasation metastasis to distinct tissues, such as the brain, kidney, lung, liver, and bone.

**Figure 7 biomedicines-09-01265-f007:**
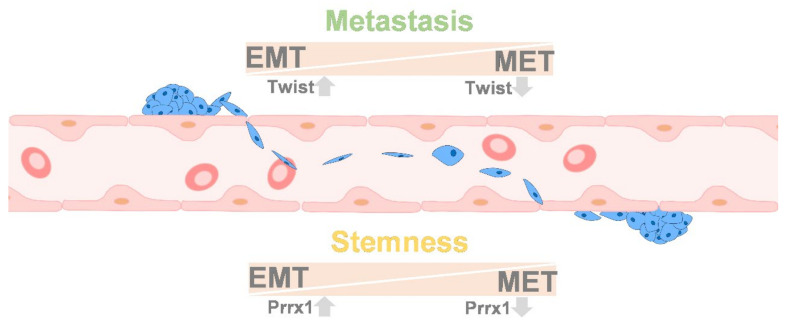
The theory of cancer stem cells. Changes in Twist and Prrx1 between EMT and MET explain the possible conversions between cell type and function. When overexpressed tumor cells twist, cells with more EMT phenotypes and undergo cell motility; once cells metastasize to the proper location, the related twist will be downregulated, and the MET process enables cell colonization. However, such a process was distinct to the cell’s stemness ability. Instead, the expression of Prxx1 could be activated by EMT but suppressed stemness. Moreover, the MET process ability stemness activity increased in colonization, in which Prxx1 was downregulated. Up arrow means upregulation. Down arrow means downregulation.

**Figure 8 biomedicines-09-01265-f008:**
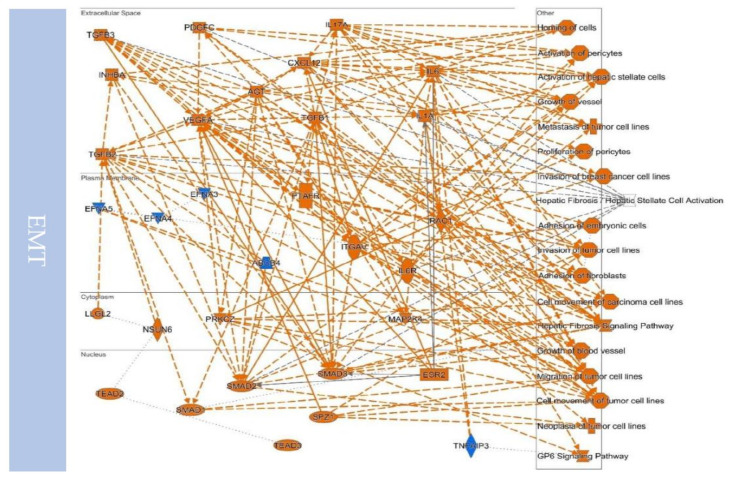
Simulated molecular network between EMT and MET. The simulation of molecular interactions between EMT or MET hallmarks and the related biological functions of participation was conducted through prediction methods, in which gene ontology is linked to cytoskeleton remodeling, metastasis, remodeling the microenvironment, and embryogenesis-related process. (MET- or EMT-related molecules were downloaded from GSEA datasets, and the analyzed interaction map results were output from the Ingenuity Pathway Analysis.).

**Figure 9 biomedicines-09-01265-f009:**
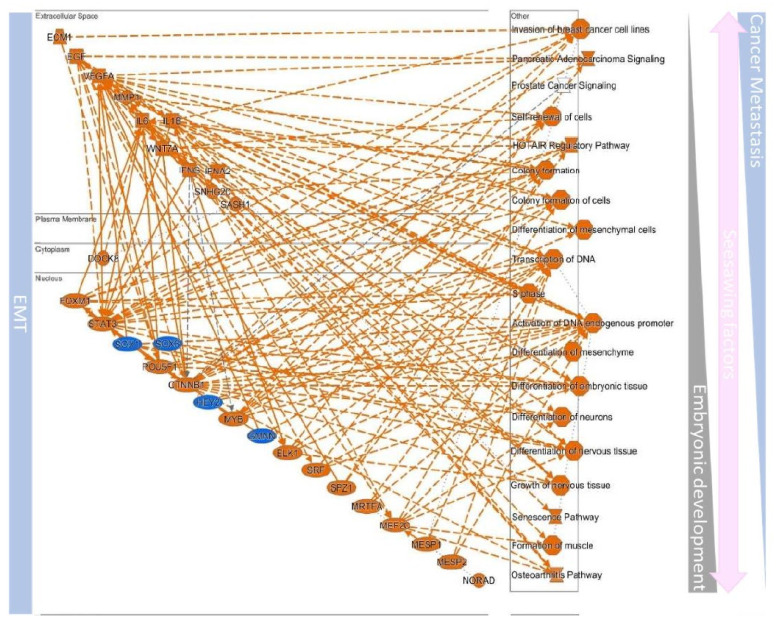
The upstream regulators of EMT/MET dominate both embryonic development and cancer progression. A simulation of molecular interactions of EMT and MET hallmarks upstream regulators and the related biological functions of participation was conducted through prediction methods.

**Figure 10 biomedicines-09-01265-f010:**
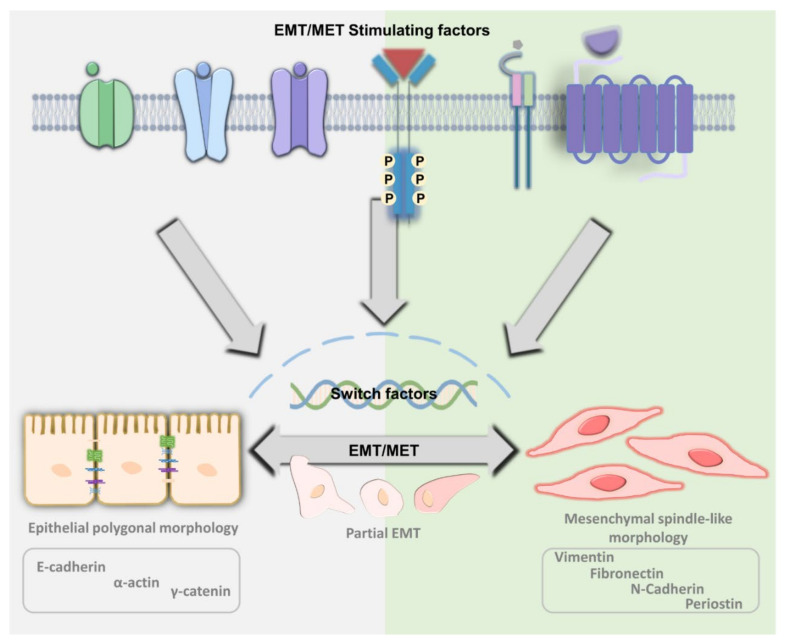
The molecular mechanism of EMT/MET. To date, multiple essential embryogenesis factors have been proven to contribute to EMT/MET by control diverse switch factor activity, in which the epithelial or mesenchymal status has been well characterized, but among them, the intermediated status, partial EMT empower cancer stemness, drug-resistant, and collective motility, such plasticity contribute to tumor heterogeneity of tumors.

**Table 1 biomedicines-09-01265-t001:** EMT- and MET-related markers. Currently, multiple markers are used to distinguish the process of cells undergoing EMT or MET, and these specific signatures can be recognized with related gene expression, transcriptional factor activity, and cell morphology.

EMT			MET		
Gene Increased	Transcriptional Factor Increased	Morphology	Gene Increased	Transcriptional Factor Increased	Morphology
*OPN* *vimentin* *matrix* *metallopro* *teinase2*	*Snail* *Slug* *Smuc* *ZEB1/2* *Twist1/2* *CDH1* *FoxC2* *TCF4* *XBP1*	Spindle-like	*E-cadherin* *Tight* *junction* *proteins* *Claudin-4*	*Ets-1* *Pax* *family* *members* *HIF1-* *α*	Epithelium like

Note: The table does not contain the noncoding RNA, protein kinase.

**Table 2 biomedicines-09-01265-t002:** EMT and MET hallmarks involved in the canonical pathway. Molecules participated in EMT or MET processes, and specific canonical pathways were found to cause a possible molecular interaction.

	Canonical Pathways	Related Factors
EMT	Hepatic Fibrosis/Hepatic Stellate Cell Activation	*ACTA2, CCN2, COL11A1, COL12A1, COL16A1, COL1A1, COL1A2, COL3A1, COL4A1, COL4A2, COL5A1, COL5A2, COL5A3, COL6A2, COL6A3, COL7A1, COL8A2, CXCL8, FAS, FGF2, FN1, IGFBP3, IGFBP4, IL6, LAMA1, MMP1, MMP2, MYL9, PDGFRB, SERPINE1, TGFB1, TIMP1, TNFRSF11B, VCAM1, VEGFA, VEGFC*
	GP6 Signaling Pathway	*COL11A1, COL12A1, COL16A1, COL1A1, COL1A2, COL3A1, COL4A1, COL4A2, COL5A1, COL5A2, COL5A3, COL6A2, COL6A3, COL7A1, COL8A2, ITGB3, LAMA1, LAMA2, LAMA3, LAMC1, LAMC2*
	Hepatic Fibrosis Signaling Pathway	*ACTA2, CCN2, COL1A1, COL1A2, COL3A1, COL5A3, CXCL8, FGF2, FZD8, ITGA2, ITGA5, ITGAV, ITGB1, ITGB3, ITGB5, JUN, LRP1, MMP1, MYL9, MYLK, PDGFRB, RHOB, SERPINE1, SPP1, TGFB1, TGFBR3, TIMP1, TNFRSF11B, VCAM1, VEGFA, VEGFC, WNT5A*
	Tumor Microenvironment Pathway	*CD44, COL1A1, COL1A2, COL3A1, CXCL12, CXCL8, FAS, FGF2, FN1, IL6, ITGA5, ITGB3, JUN, MMP1, MMP14, MMP2, MMP3, SPP1, TGFB1, TNC, VEGFA, VEGFC*
	Inhibition of Matrix Metalloproteases	*ADAM12, LRP1, MMP1, MMP14, MMP2, MMP3, SDC1, TFPI2, THBS2, TIMP1, TIMP3*
	Axonal Guidance Signaling	*ADAM12, BDNF, BMP1, CXCL12, FZD8, ITGA2, ITGA5, ITGAV, ITGB1, ITGB3, ITGB5, MMP1, MMP14, MMP2, MMP3, MYL9, PFN2, SLIT2, SLIT3, VEGFA, VEGFC, WIPF1, WNT5A*
	Leukocyte Extravasation Signaling	*ACTA2, CD44, CXCL12, EDIL3, ITGA2, ITGB1, MMP1, MMP14, MMP2, MMP3, THY1, TIMP1, TIMP3, VCAM1, WIPF1*
	Regulation of Actin-based Motility by Rho	*ACTA2, ITGA2, ITGA5, ITGAV, ITGB1, ITGB3, ITGB5, MYL9, MYLK, PFN2, RHOB, WIPF1*
	Regulation Of The Epithelial Mesenchymal Transition By Growth Factors Pathway	*CDH2, FGF2, FOXC2, ID2, IL6, JUN, MEST, MMP1, MMP2, PDGFRB, SNAI2, TGFB1, TNFRSF11B, VIM*
	Signaling by Rho Family GTPases	*ACTA2, CDH11, CDH2, CDH6, ITGA2, ITGA5, ITGAV, ITGB1, ITGB3, ITGB5, JUN, MYL9, MYLK, RHOB, VIM, WIPF1*
	ILK Signaling	*ACTA2, FERMT2, FLNA, FN1, ITGB1, ITGB3, ITGB5, JUN, MYL9, RHOB, SNAI2, VEGFA, VEGFC, VIM*
	Role of Tissue Factor in Cancer	*CCN1, CCN2, CXCL1, CXCL8, ITGAV, ITGB1, ITGB3, MMP1, PLAUR, VEGFA, VEGFC*
	HIF1α Signaling	*FGF2, IL6, JUN, MMP1, MMP14, MMP2, MMP3, SAT1, SERPINE1, TGFB1, VEGFA, VEGFC, VIM*
	Wnt/β-catenin Signaling	*CD44, CDH2, DKK1, FZD8, GJA1, JUN, LRP1, SFRP1, SFRP4, TGFB1, TGFBR3, WNT5A*
	RhoGDI Signaling	*ACTA2, CD44, CDH11, CDH2, CDH6, ITGA2, ITGA5, ITGAV, ITGB1, ITGB3, ITGB5, MYL9, RHOB*
	Regulation of the Epithelial-Mesenchymal Transition Pathway	*CDH2, FGF2, FOXC2, FZD8, ID2, LOX, MMP2, NOTCH2, PDGFRB, SNAI2, TGFB1, WNT5A*
	Glioma Invasiveness Signaling	*CD44, ITGAV, ITGB3, MMP2, PLAUR, RHOB, TIMP1, TIMP3*
	Actin Cytoskeleton Signaling	*ACTA2, FGF2, FLNA, FN1, ITGA2, ITGA5, ITGAV, ITGB1, ITGB3, ITGB5, MYL9, MYLK, PFN2*
	Integrin Signaling	*ACTA2, ITGA2, ITGA5, ITGAV, ITGB1, ITGB3, ITGB5, MYL9, MYLK, PFN2, RHOB, WIPF1*
	Colorectal Cancer Metastasis Signaling	*FZD8, IL6, JUN, LRP1, MMP1, MMP14, MMP2, MMP3, RHOB, TGFB1, VEGFA, VEGFC, WNT5A*
	IL-17 Signaling	*CXCL1, CXCL8, IL15, IL6, JUN, MMP2, MMP3, TGFB1, TNFRSF11B, VEGFA, VEGFC*
	Bladder Cancer Signaling	*CXCL8, FGF2, MMP1, MMP14, MMP2, MMP3, THBS1, VEGFA, VEGFC*
MET	Regulation of the Epithelial-Mesenchymal Transition Pathway	*FGF10, FGF7, FZD7, HGF, SMO, WNT2B, WNT4, WNT5A, WNT9B*
	Role of NANOG in Mammalian Embryonic Stem Cell Pluripotency	*BMP4, FZD7, LIF, SMO, WNT2B, WNT4, WNT5A, WNT9B*
	Basal Cell Carcinoma Signaling	*BMP4, FZD7, SMO, WNT2B, WNT4, WNT5A, WNT9B*
	Factors Promoting Cardiogenesis in Vertebrates	*BMP4, FZD7, SMO, WNT2B, WNT4, WNT5A, WNT9B*
	Human Embryonic Stem Cell Pluripotency	*BMP4, FZD7, SMO, WNT2B, WNT4, WNT5A, WNT9B*
	Regulation Of The Epithelial Mesenchymal Transition In Development Pathway	*FZD7, SMO, WNT2B, WNT4, WNT5A, WNT9B*
	Colorectal Cancer Metastasis Signaling	*FZD7, SMO, STAT1, WNT2B, WNT4, WNT5A, WNT9B*
	Ovarian Cancer Signaling	*FZD7, SMO, WNT2B, WNT4, WNT5A, WNT9B*
	Glioblastoma Multiforme Signaling	*FZD7, SMO, WNT2B, WNT4, WNT5A, WNT9B*
	WNT/β-catenin Signaling	*FZD7, SMO, WNT2B, WNT4, WNT5A, WNT9B*
	Molecular Mechanisms of Cancer	*BMP4, FZD7, SMO, WNT2B, WNT4, WNT5A, WNT9B*
	Axonal Guidance Signaling	*BMP4, FZD7, SMO, WNT2B, WNT4, WNT5A, WNT9B*
	Mouse Embryonic Stem Cell Pluripotency	*BMP4, FZD7, LIF, SMO*
	Hepatic Fibrosis Signaling Pathway	*FZD7, SMO, WNT2B, WNT4, WNT5A, WNT9B*
	Adipogenesis pathway	*BMP4, FZD7, SMO, WNT5A*
	Tumor Microenvironment Pathway	*FGF10, FGF7, HGF, TNC*

Note: The EMT hallmark was downloaded from the GSEA websites, and the molecular network was analyzed by Ingenuity Pathway Analysis.

**Table 3 biomedicines-09-01265-t003:** Upstream regulators of both EMT and MET. Transcription factors involved in EMT, MET, or EMT/MET processes were listed as switch factors. The identified upstream regulators were based on molecular interactions, as shown in Table 2 and Figure 8.

**EMT**	*VDR, CREB3L1, Pou3f1, FEZF1, YBX1, PAX7, BTG2, MEF2D, SOX7, SALL1, SP7, KDM5B, FOSL2, PPP1R13L, TAF4B, MTA2, CITED2, EP300, NKX2-5, MITF, ID3, MEN1, RFX5, ZEB1, MED4, CARM1, ERG, HLTF, TAF6, RCOR1, SMAD2, GSC, FOXQ1, PML, CBFB, BATF2, ZNF580, IRF1, NOSTRIN, RBCK1, MAX, SOX4, HOXA1, GATA3, BRCA2, PAX4, NFIA, TEAD1, HHEX, TSC22D1, HDAC4, BRCA1, ELK1, RUNX3, MECP2, SP100, NFATC4, PDLIM1, ETV3, HIC1, SQSTM1, MTA3, FOXK2, FEV, NFATC1, FOXO4, HES1, ID4, ATF3, TBX5, FOXP3, SPI1, ZNF581, NFIB, RBM14, MDM2, HOXB4, TBXT, LHX4, MAML1, BTG1, NCOR2, CBL, IKZF1, HIF3A, MED7, MIB2, RING1, TFAP4, IRF2BP1, HOXD10, BRMS1, ZFP36L1, FOXM1, ZBTB48, NKX3-1, EZH2, JMY, HTT, HES6, HDAC5, FOXA1, CREB1, HDAC7, PDX1, NEUROG3, FOXA2, SMAD1, ARNT, DDIT3, LCOR, POU2AF1, FOS, PLAG1, MXD4, JUNB, ASCC1, SMARCA4, TSHZ3, FOXL2, ZNF300, BRD7, NFKB2, TEAD3, HTATIP2, MEOX2, STAT5B, ID1, IRF5, ATF2, TFAP2C, SKI, ELF3, KLF17, KLF5, MEF2A, BARX2, APBB1, BORCS8-MEF2B, Gm21596/Hmgb1, ZNF24, ESRRA, IRF3, CEBPG, H2AX, E2F1, TSC22D3, EBF2, SP2, NRIP1, ALX4, ETS2, PMF1/PMF1-BGLAP, DAXX, EHMT2, Ncoa6, ATF1, MTDH, WTIP, ARID5B, TCF12, GLIS2, Yap1, WBP2, MYCN, TBX18, PSMD10, NFIL3, MXD3, TFAP2B, IRF6, CIITA, NUPR1, HIF1A, MED21, LPXN, HOXA4, DTX1, MYBL2, BHLHE40, ID2, TCF4, TRIM28, RB1, MZF1, STAT5A, ZNF350, HOXA9, NFYA, ZNF410, MESP1, ZEB2, CYLD, GRHL2, HEY1, DLX4, XBP1, CEBPA, TFAP2A, PBX1, IRF2, NFKBIB, RUNX2, ERF, ASH1L, BATF3, SALL4, MAFB, NFE2L2, NOTCH4, KLF6, NRF1, HEXIM1, MECOM, E2F2, HDAC3, BHLHE22, TWIST2, ZFP64, IFI16, TFEC, HOXD3, ATF6, PROX1, NFYC, TCF3, SSRP1, NEUROG1, NCOA3, WDR5, BACH1, HOXB5, SOX9, SIM1, TGIF1, SATB2, REL, CEBPD, TEAD2, HOXB8, JUND, ZFP36, PHB2, PURB, SUB1, PRDM5, TCF21, RELB, EGR2, ZNF281, SMARCA5, HNF1B, WWC1, MED24, SRA1, CREBBP, ZNF384, SMAD7, FHL2, TCF7L2, ZBTB16, ATXN1, RFX1, CALR, HEY2, ETV5, HOXB9, NAB2, SIM2, GATA2, NFKBIZ, HAND2, MAF, TAF9, ZMIZ2, ETV4, FOXP1, Cux1, ATN1, TCF20, HOXC5, NFYB, Hmgb1, EBF3, RBM39, NPAS4, CUX1, YAP1, NFKBID, BMI1, RAD21, LDB1, SOX17, FOSB, ELF4, BCL3, ZKSCAN3, CARF, CCND1, SMARCA2, NOTCH2, ZNF613, HOXC8, ZNF740, TOX, SKIL, MKX, KLF12, TGFB1I1, ARRB1, SIN3A, ECSIT, ZBTB7B, FLI1, CRTC1, CEBPZ, ZNF750, SREBF1, NFATC3, EPAS1, HOXC6, ESRRB, PHB, FOXO3, TAF4, POU2F1, ELK3, TBP, RRP1B, SREBF2, STAT6, CDKN2A, UXT, ETV6, EOMES, GATAD2B, DNAJB6, TRPS1, BCL10, HMGA1, HOXB7, ETV1, SUPT16H, BCL11B, NOTCH3, KEAP1, BACH2, MED16, ZFY, FANK1, SNAI2, KLF9, MESP2, E2F3, ZBTB46, TWIST1, SP4, PAX5, POU2F2, ATF4, GPS2, FOXO1, MYBBP1A, FOSL1, RFXANK, PIAS1, MRTFB, SRF, MAZ, PTTG1, PYCARD, GABPA, SIN3B, HMGB2, VAV1, CREB3L4, EVX2, RBL1, SMAD6, MYOCD, VHL, CREM, TFAP2E, HLX, ING2, PRDM1, HDAC6, SF1, RFX2, ZNF224, HMGB1, TRIM24, MEF2C, ARNT2, DEPDC1, KLF3*
**MET**	*POU4F2, HOXD11, IKZF2, KMT2D, TBX4, CBX4, HIVEP2, HOXA2, MEOX1, OTX2, SIX5, PAX6, ZC3H8, MEIS2, PAX9, HDAC8, ZMYND8, DLX5, NEUROG2, PAX8, LHX5, GFI1B, MED12, OTX1, FUBP1, PRRX1, DLX6, ZFPM2, HMX2, MEIS1, NSD1, HOXC11, HSF4, IRX1, COMMD3-BMI1, CNOT7, TBX1, EAF1, ZFP90, TFCP2, SHOX2, ASCL1, NKX3-2, ONECUT2, BHLHE41, ARID2, ZBTB32, PTF1A, HES3, PBX3, FOXG1*
**EMT/MET**	*CLOCK, ACTN4, IRF4, SMARCD3, PKNOX2, EHF, SS18, KLF11, USF2, HOXA13, FOXC2, SP110, ATOH1, EED, NKX2-3, DACH1, NFIX, SMAD3, SCX, DUX4, FOXE1, ZNF148, SP1, RBPJ, GATA6, HOXC9, PRDM16, FOXC1, SMARCB1, SPZ1, SPDEF, MSX1, NPM1, MYF6, STAT4, HDAC2, MRTFA, FOXF1, ETS1, HNF4A, RELA, TP73, RUNX1, EGR1, HSF1, NFATC2, GFI1, MSX2, USF1, SOX1, KLF4, MYOD1, YY1, SP3, PPARGC1A, WT1, SNAI1, LEF1, PITX2, GLI2, TP53, FOXL1, WWTR1, NFKBIE, SOX3, KLF2, ELF5, HOXA10, LMO2, STAT3, GLI1, GMNN, ASXL1, PAX3, POU5F1, HDAC1, LMX1B, CTNNB1, HOXA5, NFKB1, IRF8, SIRT1, SOX11, SIX1, EBF1, HOXA11, NFAT5, NFKBIA, PCGF2, FOXF2, KLF10, GABPB1, BCL6, SMAD4, UHRF1, GLI3, VAV2, CEBPB, PAX2, NFIC, MYC, LHX2, JUN, KDM3A, Tcf7, NKX2-1, GBX2, NOTCH1, STAT1, ARID1A, SOX2, TP63, GATA4, SMAD5, EAF2, TEAD4, MYB*

Note: The upstream regulators of EMT/MET hallmarks were analyzed from the Ingenuity Pathway Analysis.

## Data Availability

To perform the simulated molecular interaction model of EMT and MET, the related EMT or MET hallmarks were downloads from GSEA websites (https://www.gsea-msigdb.org/gsea/index.jsp; accessed on 26 July 2021), including “HALLMARK_EPITHELIAL_MESENCHYMAL_TRANSITION”, “GOBP_MESENCHYMAL_EPITHELIAL_CELL_SIGNALING” “ GOBP_MESENCHYMAL_TO_EPITHELIAL_TRANSITION” “GOBP_MESENCHYMAL_TO_EPITHELIAL_TRANSITION_INVOLVED_IN_METANEPHROS_MORPHOGENESIS”, and “GOBP_REGULATION_OF_MESENCHYMAL_TO _EPITHELIAL_TRANSITION_INVOLVED_IN _METANEPHROS_MORPHOGENESIS”, then subjected these hallmarks to the Ingenuity pathway analysis (IPA) (https://digitalinsights.qiagen.com/products-overview/discovery-insights-portfolio/analysis-and-visualization/qiagen-ipa/; accessed on 26 July 2021) to generate the related graphical summary, canonical pathway, and upstream regulators. Additionally, the Venn diagram generated the identical factors between EMT and MET.

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
