# Peer review of "Stationed or Relocating: The Seesawing EMT/MET Determinants from Embryonic Development to Cancer Metastasis"

_biomedicines, 2021, doi:10.3390/biomedicines9091265_

Round 1
Reviewer 1 Report
Based on this article title, we can imagine that some determinants information on the switching from EMT into MET. Already many information are accumulated on the drivers to proceed each process (EMT and MET), but how these can be switched is not clearly addressed. Authors need to discuss on this points if they would like to use this title for this review.
Also the difference and similarity between stemness and EMT, and dedifferentiation and EMT need to be addressed in some concepts. In these aspects, hybrid EMT (or partial EMT) concept need to be introduced clearly, as this condition is already confirmed in cancer metastasis settings. If authors discuss on cancer metastasis and EMT, they also need to discuss on the collective migration of the tumors, which is also observed in non cancer settings.
Above issues are the critical points for this manuscript to be addressed.
Author Response
Reviewer: 1
Based on this article title, we can imagine that some determinants information on the switching from EMT into MET.
- Ans: We thank the Referee for the time taken reviewing our work and for the constructive comments.
Already much information are accumulated on the drivers to proceed each process (EMT and MET), but how these can be switched is not clearly addressed. Authors need to discuss on these points if they would like to use this title for this review.
- Ans: We thank the Reviewer for bringing up this important point. The original manuscript may not have been clear about the dominant switch factors involved in the EMT and MET processes. We agree with the Reviewer that it is important to show the link between switching EMT into MET, and hence to address these issues, we have conducted a similar simulated analysis by incorporating the MET hallmark as similar to the EMT simulation. Then, identified identical upstream factors as a new Table 3. Please refer to the Table 3 lines 484-488. Finally, we used Ingenuity Pathway Analysis to overview the relationship between EMT and MET as well as their gene ontology link to metastasis or embryogenesis as a new Figure 9. Please refer to the Figure. 9 lines 489 to 492. Due to new Tables and Figures (Table. 2-3, Figure. 8-9) ( Please refer to the new Table 2 lines 280 to 283; Table 3 lines 484-488; Figure 8 lines 274 to 279; Figure 9 lines 489 to 492) , we have now more clearly included new sections to describe the upstream regulators that contribute to determining the EMT/MET, especially in cancer. Please refer to the new section lines 455 to 482. Meanwhile, we have added more discussion on the possible master regulators in the revised manuscript. Please refer to line 312-314; 358-361; 389-392; 447-453. Based on our simulated data, the potential master regulator linked to literature plays a role as a transcriptional regulator for EMT/MET in embryogenesis and carcinogenesis. We hope the particular relationships between these upstream switch factors and EMT/MET match the particular relationships between these upstream switch factors our title for this review.
Also, the difference and similarity between stemness and EMT, and dedifferentiation and EMT need to be addressed in some concepts. In these aspects, hybrid EMT (or partial EMT) concept need to be introduced clearly, as this condition is already confirmed in cancer metastasis settings. If authors discuss on cancer metastasis and EMT, they also need to discuss on the collective migration of the tumors, which is also observed in non-cancer settings. Above issues are the critical points for this manuscript to be addressed.
- Ans: We thank the Reviewer for pointing out the importance of noting the relationship regarding "the difference and similarity between stemness and EMT, and dedifferentiation and EMT" and "partial-EMT." Accordingly to the Referee's statement, we have now included the new sections to provide these concepts on related issues in the revised manuscript. Please refer to the lines 144-174. Meanwhile, we have also included numerous descriptions and discussions of the relationship between "Stemness and EMT/MET" by extended sub-section based on our simulated analysis, which may strengthen the concept of partial-EMT. Please refer to the lines 493-523.
We thank the Reviewer for this important observation and related comments as well as professional review work on our manuscript. We hope that our revised manuscript is acceptable for publication in Biomedicines.

Reviewer 2 Report
The Manuscript by Hsiao et al. "Stationed or Relocating: The Seesawing EMT/MET Determinants from Embryonic Development to Cancer Metastasis" reviews the roles of EMT and the reverse procedure of MET in cancer development. Such a review would be an important contributions to the field. I have the following comments:
1) Some paragraphs don't make sense in the context they are mentioned. For example the paragraph in lines 46-53 mention development and the next paragraph (lines 54-63) mentions EMT without any context. I would suggest adding some connective sentences between such paragraphs to explain the authors conceptual framework.
2) The figure legends should provide more information of the respective images.
3) The analysis of EMT markers with Ingenuity is an interesting addtion. If possible, I would suggest that this analysis would be also interesting to be repeated for MET markers.
Author Response
Reviewer 2:
The Manuscript by Hsiao et al. "Stationed or Relocating: The Seesawing EMT/MET Determinants from Embryonic Development to Cancer Metastasis" reviews the roles of EMT and the reverse procedure of MET in cancer development. Such a review would be an important contribution to the field. I have the following comments:
- Ans: We deeply appreciate this Reviewer for their positive and insightful suggestion.
1) Some paragraphs don't make sense in the context they are mentioned. For example, the paragraph in lines 46-53 mention development and the next paragraph (lines 54-63) mentions EMT without any context. I would suggest adding some connective sentences between such paragraphs to explain the authors conceptual framework.
- Ans: We apologize for our non-sense paragraphs. We agree with the reviewers' comments. As requested by the Referee, the Introduction section be modified and extended to better connect the relationship between epithelial and mesenchymal transition in embryogenesis and carcinogenesis. In the introduction section, we have included previous relevant references from senior author's lab on the related field. Please refer to the introduction lines 34 to 69.
2) The figure legends should provide more information of the respective images.
- Ans: We appreciate the Reviewer for raising the valuable questions. We have now included numerous descriptions of the individual images.
Please refer to the Figure 1. lines 90 to 95;
Figure 2. lines 111 to 120;
Figure 3. lines 138 to 143;
Figure 4. lines 183 to 187;
Figure 5. lines 202 to 206;
Figure 6. lines 219 to 223;
Figure 7. lines 247 to 253;
Figure 8. lines 274 to 279;
Figure 9. lines 489 to 492;
Figure 10. lines 534 to 538;
Table 1. lines 255 to 259;
Table 2. lines 280 to 283;
Table 3 lines 484 to 488.
3) The analysis of EMT markers with Ingenuity is an interesting addition. If possible, I would suggest that this analysis would be also interesting to be repeated for MET markers.
- Ans: We thank this Reviewer for their positive and constructive suggestion. We agree with the reviewers' comments. As requested by the Referee, we have conducted a similar simulated analysis by incorporating the MET hallmark similar to the EMT simulation. Interestingly, we found multiple identical gene ontologies linked to EMT and MET in embryogenesis and carcinogenesis. Please refer to the new Table 2 lines 280 to 283; Table 3 lines 484-489; Figure 8 lines 274 to 279; Figure 9 lines 489 to 492. Therefore, we reorganized the related Tables and Figures as possible. Meanwhile, we have also included numerous descriptions based on our simulated results. Please refer to line 312-314; 358-361; 389-392; 447-453; 455-482; 493-523.
We thank the Reviewer for this important observation and related comments as well as professional review work on our manuscript. We hope that our revised manuscript is acceptable for publication in Biomedicines.

Reviewer 3 Report
The review entitled “Stationed or Relocating: The Seesawing EMT/MET Determinants from Embryonic Development to Cancer Metastasis, written by Li et al., describes the mechanisms of epithelial and mesenchymal transition. The authors first elaborated the EMT process in embryo development and later discussed the EMT process during tumorigenesis. The cells, once transformed into tumor cells, assimilate the characteristic of immortality. Local growth of cells without growth restriction creates solid tumors. When cancer failed to attain enough nutrition in situ, the tumor cells will undergo EMT and invade the basal membrane of nearby blood vessels. Upon transportation through the bloodstream to secondary sites, tumor cells form colonies and undergo reverse EMT to mesenchymal-epithelial transition (MET). This vigorous changes implicated in EMT and MET are cell morphology, environmental conditions, and external stimuli. Overall, the focuses dissecting the similarities and differences between EMT and MET from embryonic development to the stage of cancer metastasis. This is a compressive review. The authors have discussed extensive research published in the area of EMT. However, one concern is that the work coming out of the senior author’s lab is not highlighted.
Author Response
Reviewer 3:
The review entitled "Stationed or Relocating: The Seesawing EMT/MET Determinants from Embryonic Development to Cancer Metastasis, written by Li et al., describes the mechanisms of epithelial and mesenchymal transition. The authors first elaborated the EMT process in embryo development and later discussed the EMT process during tumorigenesis. The cells, once transformed into tumor cells, assimilate the characteristic of immortality. Local growth of cells without growth restriction creates solid tumors. When cancer failed to attain enough nutrition in situ, the tumor cells will undergo EMT and invade the basal membrane of nearby blood vessels. Upon transportation through the bloodstream to secondary sites, tumor cells form colonies and undergo reverse EMT to mesenchymal-epithelial transition (MET). These vigorous changes implicated in EMT and MET are cell morphology, environmental conditions, and external stimuli. Overall, the focuses dissecting the similarities and differences between EMT and MET from embryonic development to the stage of cancer metastasis. This is a compressive review. The authors have discussed extensive research published in the area of EMT.
- Ans: We thank this Reviewer for their positive and constructive review.
However, one concern is that the work coming out of the senior author's lab is not highlighted.
- Ans: We thank the Reviewer for pointing this out to us. To address the Reviewer's concern, we have now included numerous to highlight the related works of the senior author's lab.
Please refer to the (yellow labels) lines 50;
line 53;
line 58;
line 66;
line 73;
line 79;
line 107;
line 110;
line 130;
line 263;
line 264;
line 265;
line 286;
line 289;
line 304;
line 306;
line 348;
line 350;
line 367;
line 373;
line 377;
line 378;
line 397;
line 398;
line 401;
line 407;
line 412;
line 413;
line 418;
line 421;
line 426;
line 433;
We appreciate the opportunity to clarify and improve our manuscript and believe we have addressed all of the comments raised by the referees. We hope you and the referees will find our revised manuscript acceptable and the paper meets the editorial requirements. Your consideration of this manuscript is greatly appreciated.
We look forward to hearing from you.

Round 2
Reviewer 1 Report
This manuscript is much improved and could be published with the minor changes listed below;
- (Figure 10) The signaling pathways shown in this figure are all stimulators for EMT (not MET), and the switch factors indicated in this figures are all the inducers (and signal transducers or TFs) for EMT (not MET). Therefore this figure could mislead readers. Please correct these points.
- GRHL2 and Ovol2 are the MET TFs, which are essential to cause MET, and surprisingly these factors are not involved in this manuscript. If authors need to discuss on EMT/MET, EMT TFs and MET TFs are essential to drive the processes. Should state on these.
- (Table 2,3, Figure 8,9) For these bioinformatical information, authors need to explain shortly (in figure legends) how these data are created. Also character quality in the figure is not good enough.
Author Response
Reviewer: 1
This manuscript is much improved and could be published with the minor changes listed below;.
- Ans: First of all, we would like to thank the Referee for the thorough examination of our revised manuscript. As far as possible, we answered to the questions raised.
1) (Figure 10) The signaling pathways shown in this Figure are all stimulators for EMT (not MET), and the switch factors indicated in these figures are all the inducers (and signal transducers or TFs) for EMT (not MET). Therefore, this Figure could mislead readers. Please correct these points.
- Ans: We apologize for these mistakes and have modified the Figure 10. We thank the Reviewer for this critical observation. We agree with the Reviewer that this Figure may mislead readers. In addition, considering the switch factors may present as dynamic states. Hence, we have removed the controversial molecules or characters to address these issues and avoid the possible dual factors between EMT and MET.
Please refer to the Figure. 10 lines 547 to 551.
2) GRHL2 and Ovol2 are the MET TFs, which are essential to cause MET, and surprisingly these factors are not involved in this manuscript. If authors need to discuss on EMT/MET, EMT TFs and MET TFs are essential to drive the processes. Should state on these.
- Ans: In the first revision round, referee2 asked us to conduct the simulated MET network as EMT, what we have done. And here, we need to clarify that all these hallmark molecules were downloaded from GSEA datasets and analyzed from IPA without any selected criteria. It is just to provide a possible simulated interaction to illustrate how these molecules are involved in the EMT and MET and contribute to embryogenesis or carcinogenesis. We confirmed these output lists and did not see the GRHL2 and Ovol2. Therefore, we apologize did not mention how important there are. We thank the Reviewer for bringing up this important point. We agree with the Reviewer that it is essential to emphasize the role of GRHL2 and OVOL2 link between switching the EMT or MET, and hence to address these issues, we have now more clearly included new states to describe the roles of GRHL2 and Ovol2 that contribute to determining the EMT/MET, especially in cancer. Please refer to the lines 470 to 488.
3) (Table 2,3, Figure 8,9) For these bioinformatical information, authors need to explain shortly (in figure legends) how these data are created. Also character quality in the Figure is not good enough.
- Ans: We thank the Reviewer for pointing out the importance. We agree with the reviewers' comments. As requested by the Referee, we have described how these data (Table 2,3, Figure 8,9) were created.
Please refer to the Resource section lines 559 to 571. - For the character quality issues, We apologize for our small characters. We agree with the reviewers' comments. As requested by the Referee, we have now replaced Tables or Figures with as enlarged characters as possible.
Please refer to the Table.2 lines 280 to 283;
Please refer to the Table.3 lines 498 to 501;
Please refer to the Figure.8 lines 275 to 279;
Please refer to the Figure.9 lines 502 to 505;
We thank the Reviewer for this critical observation and related comments as well as professional review work on our manuscript. We hope that our revised manuscript is acceptable for publication in Biomedicines.

Reviewer 2 Report
The review has been substantially improved and the authors succesfully adressed all of my concerns. Therefore, I suggest the publication of this interesting review in its present form,
Author Response
Reviewer 2:
The review has been substantially improved and the authors successfully addressed all of my concerns. Therefore, I suggest the publication of this interesting review in its present form,
- Ans: We thank this Reviewer for the positive evaluation of our manuscript.
We appreciate the opportunity to clarify and improve our manuscript and believe we have addressed all of the comments raised by the referees. We hope you and the referees will find our revised manuscript acceptable and the paper meets the editorial requirements. Your consideration of this manuscript is greatly appreciated. We look forward to hearing from you.
